# *γ*-Cyclodextrin-Encapsulated Cinnamaldehyde for Citrus Preservation and Its Potential Mechanisms against *Penicillium digitatum*

**DOI:** 10.3390/jof8111199

**Published:** 2022-11-14

**Authors:** Yonghua Zhang, Yuanzhen Tan, Okwong Oketch Reymick, Qiuli Ouyang, Nengguo Tao

**Affiliations:** School of Chemical Engineering, Xiangtan University, Xiangtan 411105, China

**Keywords:** inclusion compounds, inhibitory activity, green mold, action mechanism, membrane permeability

## Abstract

In this study, a *γ*-cyclodextrin-cinnamaldehyde inclusion compound (*γ*-CDCL) was prepared to control green mold caused by *Penicillium digitatum* (*P. digitatum*) in citrus. The results showed that the minimum inhibitory concentration (MIC) and minimum fungicidal concentration (MFC) of *γ*-CDCL against the mycelial growth of *P. digitatum* were 2.0 g L^−1^ and 4.0 g L^−1^, respectively. Simultaneously, eight × MFC *γ*-CDCL could effectively reduce the incidence of green mold in citrus fruit without impairment of the fruit qualities, meanwhile, eight × MFC *γ*-CDCL was comparable to Prochloraz in controlling fruit under natural storage conditions. The structure of *γ*-CDCL was characterized by scanning electron microscopy (SEM), X-ray diffraction (XRD), Fourier transform-infrared spectroscopy (FT-IR), and nuclear magnetic resonance (NMR) analyses. Results showed that the successful preparation of *γ*-CDCL was due to the spatial interaction between H-4,8 of cinnamaldehyde and H-5′ of *γ*-cyclodextrin. Meanwhile, the cell membrane permeability of *P. digitatum* was impaired by *γ*-CDCL through massive accumulation of reactive oxygen species, whereas the cell wall integrity was barely affected. These results indicated that *γ*-CDCL might inhibit the growth of *P. digitatum* through a membrane damage mechanism and it is a promising alternative to chemical fungicides in controlling the post-harvest citrus decay.

## 1. Introduction

Post-harvest diseases, especially green mold, caused by *Penicillium digitatum*, result in significant economic losses to post-harvest citrus crops which during storage and marketing around the world [1,2,3]. Currently, the control of citrus postharvest diseases mainly depends on the demethylation inhibitor (DMI) fungicide (such as imazalil and prochloraz), but fungicide resistance in *Penicillium digitatum* has been observed with the abuse of DMI fungicides [4,5,6]. At the same time, concerns about environmental contamination and human health threats have heightened the need to find preservatives that are environmentally friendly and can achieve the same effect as chemical fungicides [5,7].

Essential oils, which are oily liquids obtained from plants or spices, have gained the attention of people worldwide in recent years due to their superior antifungal capacity against *Geotrichum citri-aurantii*, *Penicillium digitatum*, *Botrytis cinerea*, *Penicillium italicum* and *Monilinia fructicola* [8,9,10,11,12]. Cinnamaldehyde (CL, C_9_H_8_O), is the main component of cinnamon essential oil, has a wide application in the food industry and pharmaceutical fields due to its high antifungal and anti-inflammatory activities [13]. Nevertheless, the practical application of CL is limited due to its low water solubility, high volatility, strong pungent smell and sensitivity to light and oxidation [14,15].

Encapsulation is one of the options to overcome the shortcomings of essential oils. Cyclodextrins (CDs) are natural cyclic oligosaccharides produced by enzyme-modified starch, which have a peculiar structure of a hydrophobic cavity and a hydrophilic outer surface, and can form a non-covalent inclusion compound [16,17]. Many studies have shown that plant essential oils encapsulated with cyclodextrins could increase the solubility, biodegradability, reduction in the strong odor, and maintenance of the antifungal activities of the essential oils [18,19,20]. Tea tree oil/HP-*β*-CD inclusion complexes showed superior inhibition of brown rot and were effective in reducing its disease incidence in vivo [21]. p-Anisaldehyde-*β*-cyclodextrin inclusion complexes exhibited high antioxidant activity and antifungal activity against *Escherichia coli* and *Staphylococcus aureus* [22]. The above reports indicate that cyclodextrins are an excellent essential oil delivery material [19]. As far as we know, there are three main types of cyclodextrins, namely *α*-, *β*-, and *γ*-cyclodextrins, which were formed by 6, 7, and 8 (α-1,4)-linked *α*-D-glucopyranose units, respectively [20]. In general, *γ*-CD is an ideal encapsulation material because of its larger cavity, higher water solubility, as well as excellent encapsulation ability compared to *α*-CD and *β*-CD [21,22]. Formation of perilla oil and *γ*-CD inclusion complexes has been reported to improve the thermal stability and bioavailability of perilla oil [23]. Liu et al. also investigated whether the inclusions formed between benzyl isothiocyanate and *γ*-CD had a durable antifungal effect against Staphylococcus aureus [24]. In general, complexation with *γ*-CD is a promising method that enhances the solubility and bioavailability of guest molecules [25,26].

The potential mechanisms underlying the antifungal activity of *γ*-CDCL are not yet well-established, but several possible antifungal mechanisms have been presented. For example, di-n-decyl dimethyl ammonium chloride cyclodextrin inclusion complex was dissolved in solution and adsorbed on the surface of the fungal membrane via electrostatic action, which changed the charge on the cell membrane and eventually led to cell death [27]. *β*-CD-eugenol inclusion complex was shown to have a disruptive effect on the morphology and ultrastructure of Peronophythora litchi [28]. Despite all the already completed work, still, little is known about the activity of *γ*-CDCL against *Penicillium digitatum* and the associated antifungal mechanisms.

In this study, the successful preparation of cinnamaldehyde-*γ*-cyclodextrin inclusion complex (*γ*-CDCL) was confirmed by physicochemical characterization. In addition, in vitro and in vivo experiments were performed to evaluate the effect of *γ*-CDCL in ponkan fruit that were inoculated with *Penicillium digitatum* and natural conditions. Finally, the possible antifungal mechanism of *γ*-CDCL against *Penicillium digitatum* was evaluated. In general, this study provides evidence for the development of *γ*-CDCL as an effective antifungal compound that inhibits *Penicillium digitatum* and may have promising implications for the control of green mold in citrus fruit.

## 2. Materials and Methods

### 2.1. Chemicals

Cinnamaldehyde (98%, CAS. No. 104-55-2, CL) and *γ*-cyclodextrin (98%, CAS. No. 17465-86-0, *γ*-CD) were acquired from Aladdin (Shanghai, China). Prochloraz was procured from Fumei Plant Protection Agent Co. Ltd. (Suzhou, China). Calcofluor White (CFW) was procured from Sigma-Aldrich (St. Louis, MO, USA). Membrane permeability assay kit was procured from Cablebridge Biotechnology Co., Ltd. (Shanghai, China), and the ROS assay kit was procured from Solarbio Science & Technology Co. Ltd. (Beijing, China).

### 2.2. Pathogen and Fruit

*P. digitatum* was provided by Xiangtan University, Xiangtan, China. It was propagated on potato dextrose agar (PDA) medium at 25 ± 2 °C, and the spore concentration of 5 × 10^5^ spores mL^−1^, determined using a hemocytometer, used for the subsequent in vivo experiments. Mycelia of *P. digitatum* incubated for 48 h were collected from potato dextrose broth (PDB) at 25 ± 2 °C and used for experimental analysis.

Mature ponkan fruit (*Citrus reticulata* Blanco) were harvested on 1 December 2022 from an orchard in Luxi, Hunan, China. The fruit selected for in vivo tests were uniform in size and free of scars.

### 2.3. Preparation of γ-CDCL

The *γ*-CDCL was prepared by the saturated aqueous solution method [29]. In brief, *γ*-CD (10 g) was dissolved in distilled water (25 mL) at 45 °C and stirred until completely dissolved. Then, 0.4 mL of CL was mixed with ethanol (V_CL_:V_ethanol_ = 1:4) and then added dropwise to the *γ*-CD solution stirring for 3 h. Details of the subsequent procedures are exactly as described in a previous report [29].

### 2.4. In Vitro and In Vivo Antifungal Activity of γ-CDCL

The antifungal activity of *γ*-CDCL on the growth of *P. digitatum* in vitro was assessed using the agar dilution method [30,31]. Firstly, measured amounts of *γ*-CDCL were added to each sterilized potato dextrose agar (PDA) to give desired concentrations of 0.00, 0.25, 0.50, 1.00, 2.00, 4.00, 8.00 g L^−1^. Then, twenty milliliters of the PDA was poured into sterilized Petri dishes (90 mm diameter). Finally, a 6 mm diameter disc of inoculum, taken from the edge of 7 d-old cultures of *P. digitatum*, was placed at the center of each of the Petri plates. The Petri plates were incubated at 26 ± 2 °C for 4 d. The lowest concentration that completely inhibited the growth of the pathogen after 2 or 4 d of incubation were identified as the MIC or MFC [32].

The effect of the *γ*-CDCL on *P. digitatum* in vivo was investigated as previously reported with minor modifications [30]. Two wounds (length, width = 2.5 mm, depth = 1 mm) were symmetrically punctured at the equator of the fruit with a sterile scalpel and without damaging the fruit flesh. The wounds were inoculated with 0.01 mL of *P. digitatum* spore suspension of concentration of 5 × 10^5^ cfu mL^−1^. After inoculation, the fruit were left for 4 h, then immersed in the *γ*-CDCL solution at concentrations of 4 ×, and 8 × MFC for 60 s. Prochloraz (0.025% in water, *v*/*v*) and sterile water was used as the positive control and negative control. Each treatment consisted of 3 replicates of 20 fruit and stored at 25 ± 2 °C with 85–90% relative humidity for 6 d. Fruit disease incidence was calculated and photographed, meanwhile, the physiological indicators including weight loss (WL), coloration index (CI), Firmness (N), Vitamin C (Vc) content, pH and total soluble solids (TSS) were gauged as described in previous reports [33,34].

In order to evaluate the ability of *γ*-CDCL to preserve the citrus fruit under natural storage conditions, 450 non-inoculated citrus fruit were used, of which 50 were immersed in prochloraz (0.025% in water, *v*/*v*) solution as positive control. Test groups consisted of 50 fruit were immersed in *γ*-CDCL (0 and 8 × MFC) solution for 30 s. Each experiment was repeated thrice, and the disease incidence calculated from the average values of the data collected every 10 d during the 30-day test period.

### 2.5. Characterization

The entrapment efficiency (EE) of encapsulated CL in the *γ*-CDCL was calculated by the ultrasonic-centrifugal method using a 2802S UV/VIS spectrophotometer (Unico, Shanghai, China) at 280 nm (Y = 185.38 X + 0.1935 (R^2^ = 0.999)) [30,35].

The morphology of samples (*γ*-CD and *γ*-CDCL) were visualized by scanning electron microscopy (JSM-6610LV; JEC, Japan), and their crystal structure were investigated with X-ray diffraction (XRD) analysis in an X-Ray Polycrystalline diffractometer (/D/MAX-2500/PC XRD). The Fourier transform infrared (FT-IR) spectrometry (NICOLET 380; Nicolites, Cambridge, MA, USA) was determined using the KBr pellet technique. In addition, nuclear magnetic resonance (NMR) analysis was recorded with an AVANCE III HD 400 MHz spectrometer using the procedure proposed by Dou et al. [30].

### 2.6. Inhibitory Mechanism of γ-CDCL against P. digitatum

#### 2.6.1. Scanning Electron Microscopy (SEM)

The 48 h old mycelia were treated with *γ*-CDCL (0, 1/2 MIC and MIC) for 0, 30, 60 min. Samples were then prepared for SEM observation following a previously described method [36].

#### 2.6.2. Cell Wall Integrity

Mycelia samples treated with *γ*-CDCL (see Section 2.6.1) were stained with calcofluor white (CFW) (Sigma, St. Louis, MO, USA), and examined under a fluorescent microscope. Extracellular AKP activity was examined using the AKP kit (Solarbio Science and Technology Co., Ltd., Beijing, China) according to the manufacturer’s instructions. The methods used for these two experiments are mentioned in previous reports [36,37].

#### 2.6.3. Membrane Integrity Assays

*γ*-CDCL treated mycelial samples (Section 2.6.1) were stained with propidium iodide (PI) as described in our previous study [32] and observed under a fluorescence microscope. Total lipid content in the *P. digitatum* cells was assessed using the phospho-vanillin method as explained by Wang et al. [38].

#### 2.6.4. Reactive Oxygen Species (ROS) Levels

The accumulation of intracellular ROS in *P. digitatum* mycelia treated with *γ*-CDCL (Section 2.6.1) was determined using an ROS assay kit (Solarbio, Beijing, China) following the manufacturer’s instructions [37].

#### 2.6.5. Mitochondrial Membrane Potential (MMP) Levels

The mitochondrial membrane potential (MMP) of *P. digitatum* mycelia treated with *γ*-CDCL (Section 2.6.1) was determined using the JC-10 Assay Kit (Solarbio, Shanghai, China) as directed by the manufacturer [37].

### 2.7. Statistical Analysis

All assays were repeated thrice and presented as the mean ± standard deviation. Graphs were prepared using Origin 2017 (OriginLab, Northampton, MA, USA). Data were analyzed using SPSS 16.0 software (SPSS Inc., Chicago, IL, USA) and Duncan’s multiple range test used to ascertain the differences between the means (*p* < 0.05).

## 3. Results

### 3.1. In Vitro and In Vivo Antifungal Effect of γ-CDCL

The histogram in Figure 1A indicated that γ-CDCL elicited considerable antifungal activity against *P. digitatum* as γ-CDCL concentration increased. By 2 days of incubation, 1.00 g L^−1^ of γ-CDCL showed 80.57 ± 4.88% inhibition of *P. digitatum*, and 2.00 g L^−1^ of γ-CDCL exhibited 100% inhibition of *P. digitatum* (Figure 1A). By 4 days of incubation, 2.00 g L^−1^ of γ-CDCL showed 59.17 ± 0.56% inhibition of *P. digitatum*, and 4.00 g L^−1^ of γ-CDCL exhibited 100% inhibition of *P. digitatum* (Figure 1A). Consequently, the MIC and MFC value of γ-CDCL were evaluated as 2.00 g L^−1^ and 4.00 g L^−1^.

As shown in Figure 1B, it is apparent that γ-CDCL can effectively reduce the rotting of citrus fruit inoculated with *P. digitatum*. Aside from the 8 × MFC treatment, all groups began to rot and present water stains on the surface of citrus fruit after 3 days of storage (Figure 1C). Concurrently, the incidence in fruit of the control, 4 × MFC and prochloraz treatment were 20.0%, 5.0%, and 1.7%, respectively, whereas the 8 × MFC-treated were not infected (Figure 1B). In particular, at the end of 6 days of storage, the incidence of the control group was 100% (Figure 1B), and rot and green mold were observed on the fruit surface (Figure 1C). It was interesting to note that the incidence of the 8 × MFC treated was only 31.7%, which was much lower than that of the 4 × MFC and Prochloraz-treated groups of 48.3% and 41.7% (Figure 1B).

The effects of γ-CDCL on citrus fruit quality are shown in Table 1. Throughout the storage period of inoculation, weight loss (WL), coloration index (CI), vitamin C (Vc) content, pH, and total soluble solids (TSS) increased day by day, on the contrary, firmness decreased. After 5 days of storage, no significant differences were observed in WL, CI, Vc, pH, and TSS content, but the firmness of the treatment groups were higher than the control. The firmness of the control, prochloraz, 4 × MFC and 8 × MFC-treated groups were 1.25 ± 0.02 N, 1.30 ± 0.02 N, 1.32 ± 0.01 N, and 1.32 ± 0.01 N, respectively (*p* < 0.05), which indicated that γ-CDCL can effectively maintain the hardness of citrus fruit, and that may be a beneficial phenomenon.

Evaluation of γ-CDCL on fruit under natural storage conditions is shown in Figure 1D. After 10 days of storage, there was no decay observed in 8 × MFC γ-CDCL-treated; on the contrary, the control and Prochloraz treatment groups began to decay, with an incidence of 2.00% and 0.67%, respectively. With the extension of time, after 30 days, the disease prevalence of control and prochloraz-treated groups were 22.67% and 10.67%, respectively, which were higher levels than that of the 8 × MFC γ-CDCL-treated and only performed 6.67% (*p* < 0.05). Based on these results, γ-CDCL could inhibit the incidence of green mold in citrus fruit caused by *P. digitatum* in vivo and its effect was consistent with the commercial fungicide.

### 3.2. Characterization of the γ-CDCL

The EE of γ-CDCL was determined to be 75.63 ± 1.04% at a 3 h inclusion time, 45 °C inclusion temperature. The surface microstructure and crystal shape of γ-CD (A) and γ-CDCL (B) are depicted in Figure 2. Free γ-CD presented a different size irregular crystal, and some crystals were clumping together (Figure 2A). Besides, more uniform distribution was observed in the crystal morphology of γ-CDCL (Figure 2B), and some columnar-shaped crystals were observed, which were different from the sizes and shapes of γ-CD. XRD is a useful technology for determining complexes in powder or microcrystalline forms (Dou et al., 2018). γ-CD exhibited intense and sharp peaks at 2θ values of 6.09°, 10.15°, 12.31°, 15.33°, 16.25°, 18.58°, 21.61°, 23.33°, and 28.53°, which confirmed the crystalline nature of γ-CD (Figure 2C). In addition, the new diffraction peaks of γ-CDCL appeared at 5.84°, 7.41°, 10.47°, 11.94°, 13.98°, 15.82°, 21.9°, 23.71°, 26.63°, and 38.65°, respectively (Figure 2D). Besides, compared with γ-CD crystals, the crystallinity of γ-CDCL has clearly decreased.

FT-IR was used to confirm the formation of γ-CDCL, the spectra of γ-CD, CL, and γ-CDCL are shown in Figure 3. The spectrum of γ-CD showed characteristic peaks at 3402 cm^−1^ contributing to the hydroxyl group’s stretching vibrations (Figure 3A), and the main absorption bands were observed at 2929 cm^−1^ (C–H vibration) and 1029 cm^−1^ for the C–O group vibration. For CL, there were strong absorption bands at 1677 cm^−1^ (stretching vibration of C=O), 2816 cm^−1^, 2742 cm^−1^ (C-H stretching vibration of the aldehyde group), 3059 cm^−1^ (C-H stretching vibration of benzene ring) and ranging from 670 to 1125 cm^−1^ (C-H bending vibration of benzene ring) (Figure 3B). With regard to the γ-CDCL, the main absorption peaks of γ-CD around 3388 cm^−1^, 2931 cm^−1^, and 1029 cm^−1^ were still observed, while the stretching vibration of C=O at 1668 cm^−1^ was obviously reduced (Figure 3C), and the benzene ring of the characteristic absorption peak of the 3059 cm^−1^ (C-H stretching vibration) even disappeared.

^1^H NMR spectra of CL, γ-CD and γ-CDCL are shown in Figure 4A, the characteristic peak at 4.70 ppm referenced to the D_2_O signal. According to Figure 4B, all the γ-CD and CL peaks exist in the γ-CDCL spectrum, however, an obvious chemical shift can be observed. Firstly, it was shown that the chemical shift changes (Δδ) for H-3′ and H-5′ protons were higher than other protons of γ-CD, which were −0.07 ppm and −0.13 ppm, respectively. Secondly, for CL, the Δδ values of H-3 and H-4,8 were −0.09 ppm and −0.15 ppm, respectively (Figure 4B). In addition, the 2D ROESY spectrum of γ-CDCL showed that there was a link between H-4,8 of CL and H-5′ of γ-CD (Figure 4C). The above evidence showed that γ-CDCL might be formed by the interactions between H-4,8 of CL and H-5′ of γ-CD (Figure 4D).

### 3.3. Inhibitory Mechanism of γ-CDCL against P. digitatum

#### 3.3.1. The Mycelia Surface Morphology of *P. digitatum*

The surface morphology of *P. digitatum* after being γ-CDCL treated can be seen in Figure 5. It can be clearly seen that the surface of the treated mycelia changed significantly with the increase in treatment time and concentration, the 1/2 MIC treatment group at 60 min showed obvious collapse, distortions, and breakage (Figure 5E), while almost no change can be seen in the control group (Figure 5A,D).

#### 3.3.2. Cell Wall Integrity

The fluorescence intensity treated with *γ*-CDCL and control samples showed no remarkable changes at 30 min (Figure 6A). The same results could be observed at 60 min of 1/2 MIC *γ*-CDCL-treated and control (Figure 6A). However, with the increase in processing time and concentration, the bright blue fluorescence in the cell walls of *P. digitatum* was slightly uneven, and the bright blue fluorescence in the septa became weaker.

The effect of γ-CDCL on extracellular AKP activity of *P. digitatum* is shown in Figure 6B. After 120 min of incubation, the extracellular AKP activity of the 1/2 MIC-treated group was 5.38 ± 0.12 U/L, which was significantly higher than that of the control group (5.41 ± 0.03 U/L) (*p* < 0.05). The extracellular AKP activity under the 1/2 MIC-treated was not significantly different from that in the control group at 60 min; this demonstrated a good agreement with the fluorescence intensity results.

#### 3.3.3. Plasma Membrane Permeability Analysis

Evaluation of γ-CDCL on *P. digitatum* membrane permeability is shown in Figure 7A. No red fluorescence was observed in the control groups during the entire incubation. After 30 min incubation, the MIC-treatment group show distinct red fluorescence, while no red fluorescence was observed in the control and 1/2 MIC-treatment groups. With the extension at 60 min, the mycelium under the 1/2 MIC-treated showed red fluorescence, which may be caused by the destruction of cell membranes. The above results were in agreement with the fluorescence values presented in Figure 7B.

The effect of γ-CDCL on the total lipid content of *P. digitatum* is shown in Figure 7C. There was no variation in the total lipid content between the treatment and control groups after 30 min. After 60 min of treatment, the total lipid contents in the 1/2 MIC and MIC γ-CDCL-treated were 262.66 ± 2.30 and 234.93 ± 8.20 mg/g, respectively, which were significantly lower than that of the control group at 325.41 ± 11.27 mg/g (*p* < 0.05). This further indicated that the integrity of the cell membrane was damaged.

#### 3.3.4. ROS Levels

Evaluation of *γ*-CDCL on ROS levels of *P. digitatum* is shown in Figure 8. The fluorescence value of MIC *γ*-CDCL-treated (1.75 ± 0.24) was significantly higher than that of the control group (1.09 ± 0.21) at 30 min (Figure 8B). With the extension at 60 min, the fluorescence values of 1/2 MIC and MIC treatment groups were 1.34 ± 0.17 and 1.88 ± 0.26 (*p* < 0.05) (Figure 8B), respectively, which were higher than that of the control group (0.93 ± 0.12).

#### 3.3.5. The Effect of γ-CDCL on the MMP of *P. digitatum*

Given that ROS eruption was observed, the effect of *γ*-CDCL on the MMP was assessed (Figure 9). It can be seen that the MMP of the control remained stable overall (Figure 9A). After 30 min treatment, the fluorescence values of the 1/2 MIC and MIC-treated groups were 0.65 ± 0.05 and 0.60 ± 0.05, respectively, which were not significantly different from that of the control group (0.74 ± 0.06). After 60 min of treatment, the fluorescence values were 0.64 ± 0.03 and 0.42 ± 0.04 at the 1/2 MIC and MIC-treated groups, respectively, which were significantly different from 0.73 ± 0.06 in the control group (*p* < 0.05) (Figure 9B). Moreover, with the extension at 120 min, MMP was further decreased in the 1/2 MIC *γ*-CDCL treated group, which indicated that the mitochondrial function of *P. digitatum* was impaired (Figure 9B).

## 4. Discussion

In recent years, the development of botanical fungicides has been continuously explored; thus, the ability of essential oils to inhibit fungi is one of the reasons for the interest in the oils as a component of biological means for controlling the post-harvest decay of fruit [39,40]. As the active component in cinnamon EOs, CL has been reported to have significant antifungal activity against *P. digitatum* and *G. citri-aurantii* [10,41]. In this study, *γ*-CDCL considerably suppressed the growth of *P. digitatum* cells with an MFC value of 4.00 g L^−1^. The in vivo experiment showed that the inhibition effect of 8 × MFC *γ*-CDCL on green mold was significantly better than that of the prochloraz treatment group at the end of the storage period. This demonstrated that *γ*-CDCL is a promising candidate for controlling the occurrence of green mold in citrus fruit with comparable effects to a chemical fungicidal. With absence of inoculation, there was no significant difference between 8 × MFC *γ*-CDCL and prochloraz treatments. The above results indicate that *γ*-CDCL may have excellent application prospects in the control of post-harvest citrus diseases. Additionally, *γ*-CDCL treatment did not impair fruit quality which is consistent with the findings of Zhang et al. [31].

Encapsulation of EOs with CDs could interfere with cyclodextrin crystallization [42]. In the current study, the apparent morphology of *γ*-CDCL exhibited more homogeneous and regular crystals compared to cyclodextrins, indicating that the embedding of the guest molecule changed the surface morphology and crystal structure of the material [31]. This was also proved by the XRD and FT-IR analyses. In the FT-IR analysis, it was reflected in the characteristic peaks of the aldehyde groups including a reduction in the stretching vibration of C=O (1677 cm^−1^) and the disappearance of the stretching vibration of C-H (2816 cm^−1^, 2742 cm^−1^). It was also seen in the disappearance of the benzene ring (C-H stretching vibration at 3059 cm^−1^) in CL due to the shielding effect of *γ*-CD; similar phenomena have been reported previously [43]. Furthermore, the NMR spectra analysis proclaimed that the remarkable chemical shifts of CL in *γ*-CDCL due to the anisotropic shielding induced by the ring-current effect that the benzene ring and aldehyde group of CL included into the *γ*-CD cavity, and corresponds to the results of the FT-IR; a similar phenomenon has been mentioned in that the changes of the chemical shifts were most probably due to the presence of the guest molecule in the cavity of cyclodextrin [30,44]. The 2D ROESY NMR can provide decisive evidence to reveal the internal interactions of host–guests, especially those of the inclusion complexes [45]. The 1H NMR and 2D ROESY spectra uncovered that CL was incorporated into the cavity of *γ*-CD and that there was a spatial correlation between its H-4,8 and the H-5′ of *γ*-CD. Thus, we can deduce that the inclusion mode of the *γ*-CDCL is roughly as shown in Figure 4D. This is quite different from the encapsulation mode of thymol/*β*-cyclodextrin and the *trans*-2-hexenal/*β*-cyclodextrin inclusion complexes reported by Dou et al. and Zhang et al. [30,31]. In brief, these reports, together with our SEM, XRD, FT-IR, and NMR results, indicated that the *γ*-CDCL was successfully prepared.

Apparently, the *γ*-CDCL has a superior antifungal effect on *P. digitatum* and reduces the incidence of green mold in citrus. However, its underlying antifungal mechanism against *P. digitatum* remains largely unknown. In the current study, the surface structure of the mycelia was significantly different between the *γ*-CDCL-treated and control samples. The mycelial surface shriveled and collapsed after *γ*-CDCL treatment, while those untreated remained smooth and regular in shape. The distortion in mycelia surface morphology may result from an increase in cell permeability [46]. Calcofluor White is a specific staining agent for cell walls, which has a qualitative effect on the damaged cell wall [47]. As an essential parameter of cell-wall permeability detection, alterations at the levels of blue fluorescence in the septa and changes of extracellular AKP predicted that the cell-wall integrity was disturbed [48]. As can be noted from Figure 6A, the bright blue fluorescence had no visible difference between the control and 1/2 MIC treatment groups at 60 min, suggesting that the *γ*-CDCL did not damage the cell wall. This result was supported by the extracellular AKP activity.

Red fluorescence is often observed when the PI stain penetrates the damaged cell membranes and combines with nucleic acids [49,50,51]. In our present study, the red fluorescence emitted by mycelia was significantly enhanced by *γ*-CDCL treatment. Lipids are essential components of cell membranes, and a reduction in lipid content indicates that irreversible membrane damage has occurred [9,52]. The total lipid content in our study was significantly reduced in the 1/2 MIC *γ*-CDCL-treated samples, and together with the emission of red fluorescence, they significantly show that *γ*-CDCL treatment led to damage of the cell membrane. Plasma membrane is thus an important antifungal target for *γ*-CDCL.

Mitochondria are the primary source of cellular reactive oxygen species (ROS) and they play an irreplaceable role in the growth of most fungi [43,53]. The decrease in MMP is generally attributed to the undesirable accumulation of ROS, which may impact the normal morphology and function of mitochondria [54]. In the present study, an ROS burst and a decrease in mitochondrial membrane potential (MMP) were observed in *P. digitatum* after *γ*-CDCL treatment. Eugenol and citral have also been reported to inhibit the growth of fungi by reducing MMP and inducing oxidative stress, which affected the morphology and function of mitochondria [39,55]. These results have revealed that activation of the oxidative burst in *P. digitatum* cells is one of the important factors of antifungal mechanism of *γ*-CDCL. This is consistent with the study of the effect of flurochloridone in mouse Sertoli cells in which apoptosis induced by flurochloridone was mainly attributed to mitochondrial damage and cytotoxicity caused by ROS accumulation [56].

## 5. Conclusions

Firstly, *γ*-CDCL was successfully prepared with CL and *γ*-CD, which were verified by the SEM, FT-IR, XRD, and HNM. In vivo tests indicated that *γ*-CDCL can effectively reduce the incidence of citrus green mold and its effect was better than that of prochloraz treatment; additionally, similar results were observed in natural storage conditions. We have also uncovered that *γ*-CDCL exhibits its antifungal activity against *P. digitatum* by: (i) inducing accumulation of ROS; and (ii) disrupting the fungal cell membrane permeability. On the whole, this work contributes to the design of new preservative technologies that could potentially replace the chemical fungicides used in the control of post-harvest green mold in citrus.

## Figures and Tables

**Figure 1 jof-08-01199-f001:**
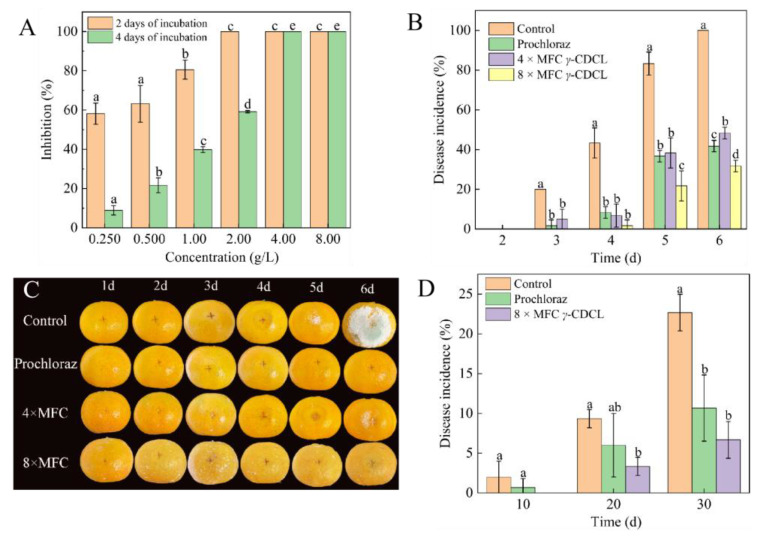
Evaluation of *γ*-CDCL on the mycelial growth of *P. digitatum* after 2 and 4 days of culture (**A**); Green mold disease incidence (**B**) and growth progression (**C**) in inoculated citrus fruit treated with *γ*-CDCL (0×, 4×, and 8× MFC) and Prochloraz (0.025% in water, *v*/*v*) during storage at 25 ± 2 °C for 6 d and 85–90% RH; Decay rate of citrus fruit under natural storage conditions for 30 days (**D**). Dissimilar lowercase letters in the histogram express the statistical significance among different treatments at the same time. Data are expressed as the mean ± standard deviation of pooled data (*n* = 3, *p* < 0.05). *γ*-CDCL, cinnamaldehyde/*γ*-cyclodextrin inclusion complex; MFC, minimum fungicidal concentration; RH, relative humidity.

**Figure 2 jof-08-01199-f002:**
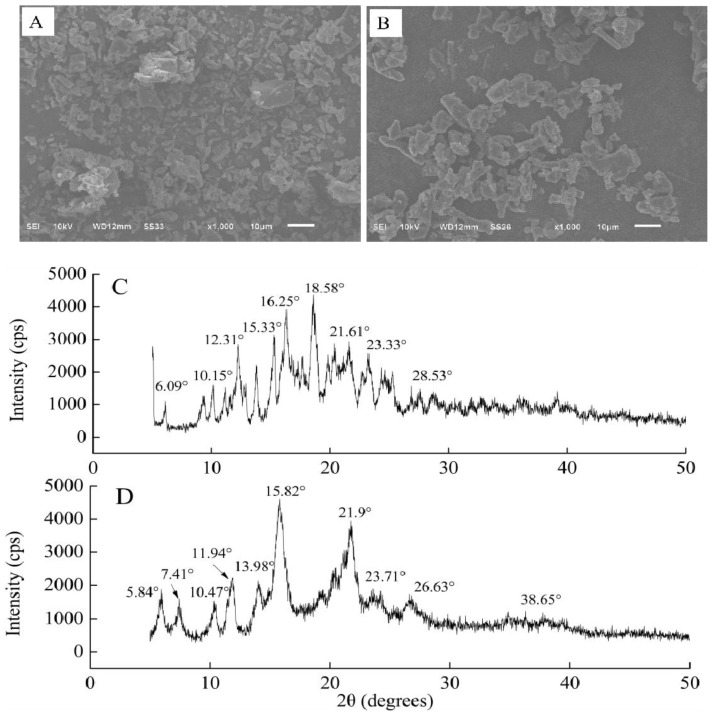
SEM graphs of *γ*-CD (**A**), *γ*-CDCL (**B**); and XRD patterns of *γ*-CD (**C**), *γ*-CDCL (**D**); SEM, scanning electron microscopy; XRD, X-ray diffraction; *γ*-CD, *γ*-cyclodextrin; *γ*-CDCL, cinnamaldehyde/*γ*-cyclodextrin inclusion complex.

**Figure 3 jof-08-01199-f003:**
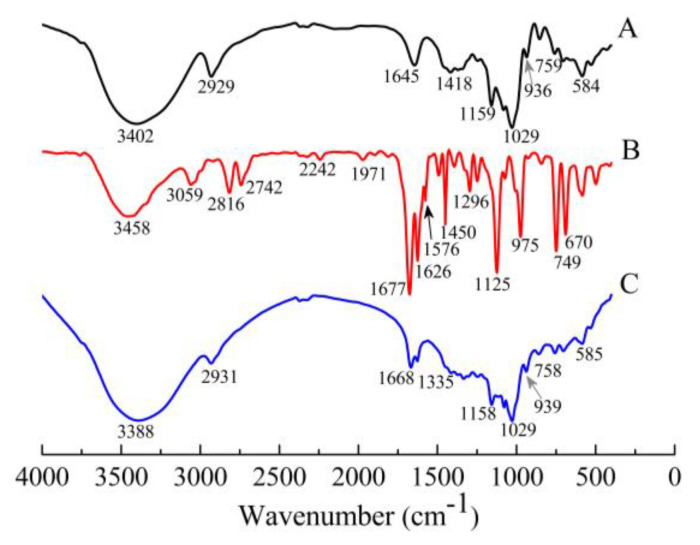
FT-IR spectra of *γ*-CD (**A**), CL (**B**), *γ*-CDCL (**C**). FT-IR, Fourier transform infrared; *γ*-CD, *γ*-cyclodextrin; CL, cinnamaldehyde; *γ*-CDCL, cinnamaldehyde/*γ*-cyclodextrin inclusion complex.

**Figure 4 jof-08-01199-f004:**
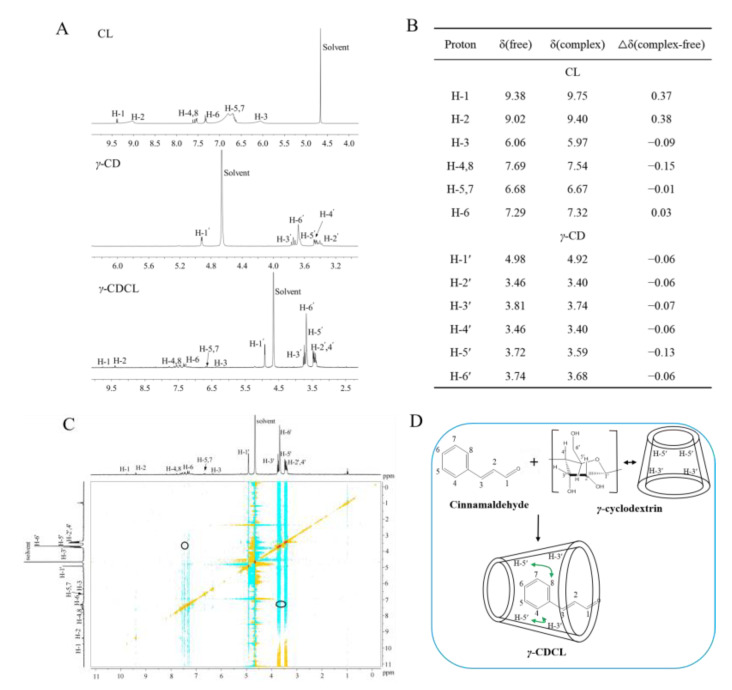
^1^H NMR spectra of CL, *γ*-CD, and *γ*-CDCL in D_2_O (**A**); Chemical shifts of protons of CL and *γ*-CD in *γ*-CDCL (**B**); The 2D ROESY of the *γ*-CDIC in D_2_O (**C**); The encapsulation mode of the *γ*-CDCL (**D**). ^1^H NMR, ^1^H nuclear magnetic resonance; 2D ROESY, two-dimensional rotating frame nuclear Overhauser spectroscopy; CL, cinnamaldehyde; *γ*-CD, *γ*-cyclodextrin; *γ*-CDCL, cinnamaldehyde/*γ*-cyclodextrin inclusion complex.

**Figure 5 jof-08-01199-f005:**
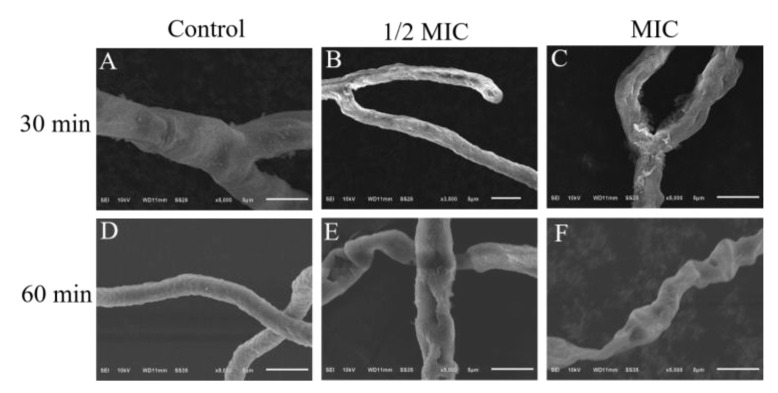
Micrographs of control samples for 30 min (**A**) and 60 min (**D**); Micrographs of *P. digitatum* hyphae treated with 1/2 MIC *γ*-CDCL for 30 min (**B**) and 60 min (**E**), or treated with 1 × MIC *γ*-CDCL for 30 min (**C**) and 60 min (**F**). SEM, scanning electron microscopy; *γ*-CDCL, cinnamaldehyde/*γ*-cyclodextrin inclusion complex; MIC, minimum inhibitory concentration.

**Figure 6 jof-08-01199-f006:**
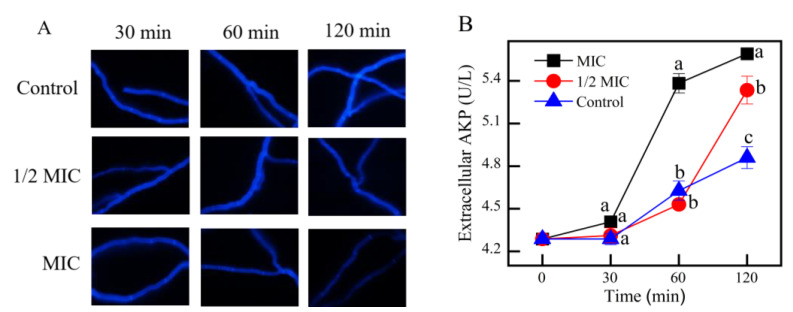
The effect of *γ*-CDCL on the cell walls of *P. digitatum.* (**A**) CFW staining images of *P. digitatum* mycelia treated with *γ*-CDCL (Control; 1/2 × MIC; 1 × MIC) for 30, 60, 120 min; (**B**) The extracellular AKP activity of *P. digitatum*. Data are expressed as the means ± standard deviation of pooled data (*n* = 3, *p* < 0.05). *γ*-CDCL, cinnamaldehyde/*γ*-cyclodextrin inclusion complex; CFW, calcofluor white; AKP, alkaline phosphatase; MIC, minimum inhibitory concentration. Dissimilar lowercase letters in the histogram express the statistical significance among different treatments at the same time.

**Figure 7 jof-08-01199-f007:**
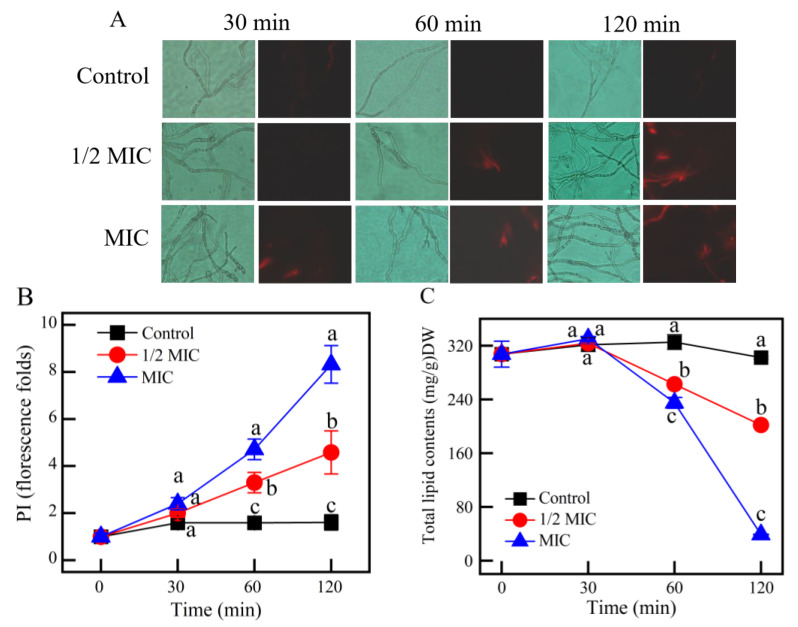
Effect of *γ*-CDCL on plasma membrane permeability of *P. digitatum* mycelia. (**A**) Fluorescence images of different treatment concentrations (Control; 1/2 × MIC; 1 × MIC) with *γ*-CDCL for 30, 60, 120 min; (**B**) The fluorescence folds change of PI staining. (**C**) Effect of *γ*-CDCL on the total lipid contents of *P. digitatum* mycelia. Data are expressed as the means ± standard deviation of pooled data (*n* = 3, *p* < 0.05). *γ*-CDCL, cinnamaldehyde/*γ*-cyclodextrin inclusion complex; MIC, minimum inhibitory concentration. Dissimilar lowercase letters in the histogram express the statistical significance among different treatments at the same time.

**Figure 8 jof-08-01199-f008:**
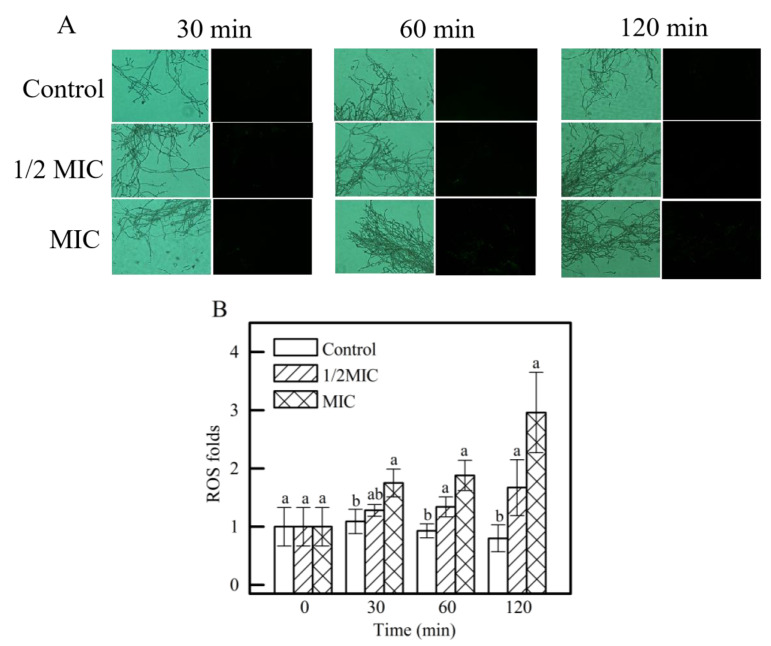
Effect of *γ*-CDCL on the ROS accumulation of *P. digitatum* mycelia. (**A**) Fluorescence images of different treatment concentrations (Control; 1/2 × MIC; 1 × MIC) for 30, 60, 120 min; (**B**) the fluorescence fold changes of the DCFH-DA staining. Data are expressed as the means ± standard deviation of pooled data (*n* = 3, *p* < 0.05). *γ*-CDCL, cinnamaldehyde/*γ*-cyclodextrin inclusion complex; ROS, reactive oxygen species; MIC, minimum inhibitory concentration; DCFH-DA, 2′−7′ dichlorofluorescein diacetate. Dissimilar lowercase letters in the histogram express the statistical significance among different treatments at the same time.

**Figure 9 jof-08-01199-f009:**
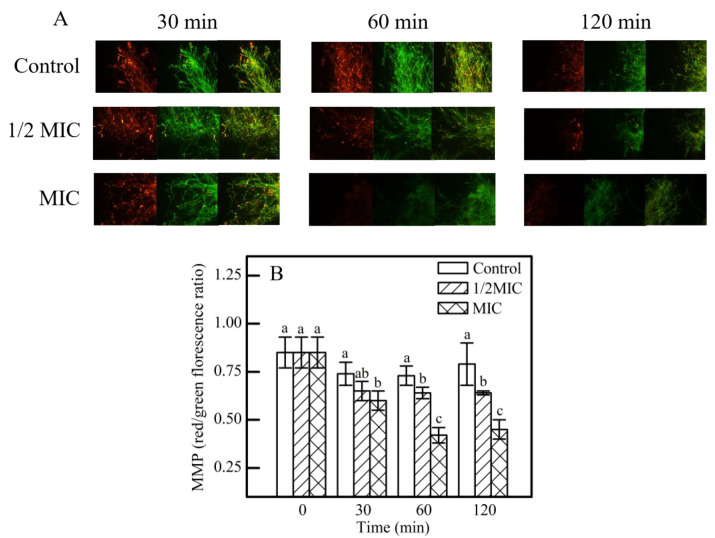
Effect of *γ*-CDCL on the MMP of *P. digitatum* mycelia. (**A**) Fluorescence images of different treatment concentrations (Control; 1/2 × MIC; 1 × MIC) for 30, 60, 120 min; (**B**) Mycelia fluorescence fold changes. Data are expressed as the means ± standard deviation of pooled data (*n* = 3, *p* < 0.05). *γ*-CDCL, cinnamaldehyde/*γ*-cyclodextrin inclusion complex; MMP, mitochondrial membrane potential; MIC, minimum inhibitory concentration. Dissimilar lowercase letters in the histogram express the statistical significance among different treatments at the same time.

**Table 1 jof-08-01199-t001:** Evaluation of the *γ*-CDCL (control, 4 × MFC and 8 × MFC) and Prochloraz treatments on post-harvest qualities of citrus fruit inoculated with *P. digitatum*.

Quality Index	Treatment	0d	1d	3d	5d
WL (%)	Control	0.00 ± 0.00 a	0.75 ± 0.11 ab	2.95 ± 0.42 ab	4.81 ± 0.52 a
	Prochloraz	0.00 ± 0.00 a	0.83 ± 0.11 a	3.30 ± 0.51 a	4.61 ± 0.64 a
	4 × MFC	0.00 ± 0.00 a	0.63 ± 0.10 b	2.28 ± 0.42 b	4.11 ± 0.40 a
	8 × MFC	0.00 ± 0.00 a	0.69 ± 0.15 ab	2.68 ±0.63 ab	4.36 ± 1.14 a
CI	Control	2.94 ± 0.72 a	2.84 ± 0.39 a	4.07 ± 0.81 a	4.24 ± 0.82 a
	Prochloraz	2.77 ± 0.62 b	2.87 ± 0.75 a	4.02 ± 1.02 a	4.10 ± 0.61 a
	4 × MFC	2.89 ± 0.66 ab	3.88 ± 0.58 a	4.14 ± 0.90 a	4.95 ± 0.52 a
	8 × MFC	2.95 ± 0.66 ab	2.85 ± 0.64 a	3.92 ± 0.43 a	4.20 ± 0.75 a
Firmness (N)	Control	1.51 ± 0.09 a	1.40 ± 0.04 a	1.30 ± 0.02 a	1.25 ± 0.02 a
	Prochloraz	1.48 ± 0.12 a	1.36 ± 0.03 a	1.31 ± 0.01 a	1.30 ± 0.02 b
	4 × MFC	1.54 ± 0.04 a	1.42 ± 0.05 a	1.31 ± 0.02 a	1.32 ± 0.01 b
	8 × MFC	1.51 ± 0.02 a	1.36 ± 0.01 a	1.39 ± 0.02 b	1.32 ± 0.01 b
	Control	13.12 ± 0.73 a	14.28 ± 2.63 a	16.80 ± 3.23 a	18.09 ± 2.56 a
Vc (mg kg^−1^)	Prochloraz	14.47 ± 1.80 a	15.12 ± 0.58 a	18.54 ± 2.50 a	19.19 ± 0.78 a
	4 × MFC	13.25 ± 0.30 a	14.73 ± 2.47 a	15.89 ± 1.66 a	19.13 ± 2.33 a
	8 × MFC	13.63 ± 2.07 a	15.83 ± 2.65 a	18.41 ± 1.97 a	18.87 ± 3.70 a
pH	Control	2.67 ± 0.06 a	3.11 ± 0.12 a	3.26 ± 0.13 a	3.29 ± 0.01 a
	Prochloraz	2.74 ± 0.03 a	3.06 ± 0.20 a	3.19 ± 0.11 a	3.27 ± 0.03 a
	4 × MFC	2.96 ± 0.18 b	3.08 ± 0.17 a	3.21 ± 0.06 a	3.29 ± 0.18 a
	8 × MFC	2.98 ± 0.01 b	3.03 ± 0.03 a	3.13 ± 0.03 a	3.28 ± 0.02 a
	Control	9.37 ± 0.25 a	9.40 ± 0.44 ab	10.24 ± 0.05 ab	10.99 ± 0.02 a
TSS	Prochloraz	9.45 ± 0.07 a	9.20 ± 0.40 b	9.71 ± 0.12 b	10.81 ± 0.05 a
	4 × MFC	9.20 ± 1.18 a	10.30 ± 0.10 a	10.43 ± 0.83 ab	10.89 ± 0.19 a
	8 × MFC	9.33 ± 0.06 a	10.33 ± 0.93 a	10.61 ± 0.29 a	11.05 ± 0.30 a

Note: Dissimilar lowercase letters in the table express the statistical significance among different treatments. Data are listed as the means ± standard deviation of pooled data (*n* = 3, *p* < 0.05). *γ*-CDCL, cinnamaldehyde/*γ*-cyclodextrin inclusion complex; MFC, minimum fungicidal concentration; WL, weight loss; CI, coloration index; Vc, Vitamin C; TSS, total soluble solids.

## Data Availability

The datasets generated during and/or analyzed during the current study are available from the corresponding author upon reasonable request.

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
