# Peer review of "γ-Cyclodextrin-Encapsulated Cinnamaldehyde for Citrus Preservation and Its Potential Mechanisms against Penicillium digitatum"

_jof, 2022, doi:10.3390/jof8111199_

Round 1

Reviewer 1 Report

In general terms the topic of the article is interesting, the methodology is explicitly presented and the results reported are interesting.

Abstract should have some result of your work

Keywords should not duplicate information from the Title of the work. This should be amended.

The introduction chapter do not abbreviation of the pathogen you should write it Penicillium

Instated of you cited old refe. 1, 2 and 3 you can use new references

and should end with a paragraph indicating the purposefulness of the conducted research. Authors should clearly define the purpose of the work and formulate research hypotheses.

Material and methods

You should write scientific name in italic Citrus reticulata

Discussion part should be separate from the result because some time authors did not discussed some part and wait till the end of the results 

Make sure that all scientific names in the References list are italics.

The paper needs some editorial corrections.

I recommend the publication of this manuscript in the Journal of Fungi journal after minor revisions.

Reviewer 2 Report

This is an interesting manuscript about the antifungal mechanisms of γ-cyclodextrin-cinnamaldehyde inclusion compound (γ-CDCL) on the fungal pathogen Penicillium digitatum, the causal agent of citrus green mold, with emphasis on the mode of action of γ-CDCL by various techniques.

The manuscript shows new results, and the objectives are apparent. The results are well presented. However, I have some points that need to be addressed as follows.

1. Line 103: Please give the required details about the effect of γ-CDCL on P. digitatum growth in vitro.

2. Lines 105-107: “The lowest concentration that completely inhibited the growth of the pathogen after the 4 d of incubation was identified as the MFC [32]”. This is MIC, not MFC, please explain this point.

In general, there is an extensive overlap in the use of both MIC and MFC throughout the whole manuscript.

3. Line 107: The full names of abbreviations should be given in the first appearance.

4. Line 122: Please mention the source of prochloraz.

5. Line 164: All assays were repeated thrice and presented as the mean ± standard deviation, and in the caption of Figure 1, and Table 1: Data are expressed as the means ± standard errors. What is correct; mean ± standard deviation (SD) or mean ± standard error (SE)?

6. The obtained results need more deep discussion.

Round 2

Reviewer 2 Report

Thanks for considering the suggestions from the previous version. The manuscript is acceptable from my point of view. I recommend accepting it for publication.